# Representational aspects of depth and conditioning in normalizing flows

## Abstract

Normalizing flows are among the most popular paradigms in generative modeling, especially for images, primarily because we can efficiently evaluate the likelihood of a data point. Training normalizing flows can be difficult because models which produce good samples typically need to be extremely deep and can often be poorly conditioned: since they are parametrized as invertible maps from $\mathbb{R}^d \to \mathbb{R}^d$, and typical training data like images intuitively is lower-dimensional, the learned maps often have Jacobians that are close to being singular. In our paper, we tackle representational aspects around depth and conditioning of normalizing flows: both for general invertible architectures, and for a particular common architecture, affine couplings. We prove that $\Theta(1)$ affine coupling layers suffice to exactly represent a permutation or $1 \times 1$ convolution, as used in GLOW, showing that representationally the choice of partition is not a bottleneck for depth. We also show that shallow affine coupling networks are universal approximators in Wasserstein distance if ill-conditioning is allowed, and experimentally investigate related phenomena involving padding. Finally, we show a depth lower bound for general flow architectures with few neurons per layer and bounded Lipschitz constant.

## 1. Introduction

Deep generative models are one of the lynchpins of unsupervised learning, underlying tasks spanning distribution learning, feature extraction and transfer learning. Parametric families of neural-network based models have been improved to the point of being able to model complex distributions like images of human faces. One paradigm that has received a lot attention is normalizing flows, which model distributions as pushforwards of a standard Gaussian

(or other simple distribution) through an *invertible* neural network $G$. Thus, the likelihood has an explicit form via the change of variables formula using the Jacobian of $G$. Training normalizing flows is challenging due to a couple of main issues. Empirically, these models seem to require a much larger size than other generative models (e.g. GANs) and most notably, a much larger depth. This makes training challenging due to vanishing/exploding gradients. A very related problem is *conditioning*, more precisely the smallest singular value of the forward map $G$. It's intuitively clear that natural images will have a low-dimensional structure, thus a close-to-singular $G$ might be needed. On the other hand, the change-of-variables formula involves the determinant of the Jacobian of $G^{-1}$, which grows larger the more singular $G$ is.

While recently, the universal approximation power of various types of invertible architectures has been studied if the input is padded with a sufficiently large number of all-0 coordinates (Dupont et al., 2019; Huang et al., 2020) or arbitrary partitions and permutations are allowed (Teshima et al., 2020), precise quantification of the cost of invertibility in terms of the depth required and the conditioning of the model has not been fleshed out.

In this paper, we study both mathematically and empirically representational aspects of depth and conditioning in normalizing flows and answer several fundamental questions.

## 2. Related Work

On the empirical side, flow models were first popularized by (Dinh et al., 2014), who introduce the NICE model and the idea of parametrizing a distribution as a sequence of transformations with triangular Jacobians, so that maximum likelihood training is tractable. Quickly thereafter, (Dinh et al., 2016) improved the affine coupling block architecture they introduced to allow non-volume-preserving (NVP) transformations, (Papamakarios et al., 2017) introduced an autoregressive version, and finally (Kingma & Dhariwal, 2018) introduced 1x1 convolutions in the architecture, which they view as relaxations of permutation matrices—intuitively, allowing learned partitions for the affine blocks. Subsequently, there have been variants on these ideas: (Grathwohl et al., 2018; Dupont et al., 2019; Behrmann et al., 2018) viewed these models as discretizations of ODEs and introduced ways to approximate determinants of non-triangular

[1]Anonymous Institution, Anonymous City, Anonymous Region, Anonymous Country. Correspondence to: Anonymous Author <anon.email@domain.com>.

Preliminary work. Under review by INNF+ 2021. Do not distribute.

Jacobians, though these models still don't scale beyond datasets the size of CIFAR10. The conditioning/invertibility of trained models was experimentally studied in (Behrmann et al., 2019), along with some "adversarial vulnerabilities" of the conditioning. Mathematically understanding the relative representational power and statistical/algorithmic implications thereof for different types of generative models is still however a very poorly understood and nascent area of study.

Most closely related to our results are the recent works of (Huang et al., 2020), (Zhang et al.) and (Teshima et al., 2020). The first two prove universal approximation results for invertible architectures (the former affine couplings, the latter neural ODEs) if the input is allowed to be padded with zeroes. The latter proves universal approximation when GLOW-style permutation layers are allowed through a construction that operates on one dimension at a time. This is very different than how flows are trained in practice, which is typically with a partition which splits the data roughly in half. It also requires the architectural modification of GLOW to work. As we'll discuss in the following section, our results prove universal approximation even without padding and permutations, but we focus on more fine-grained implications to depth and conditioning of the learned model and prove universal approximation in a setting that is used in practice. Another work (Kong & Chaudhuri, 2020) studies the representational power of Sylvester and Householder flows, normalizing flow architectures which are quite different from affine coupling networks. In particular, they prove a depth lower bound for local planar flows with bounded weights; for planar flows, our general Theorem 4 can also be applied, but the resulting lower bound instances are very different (ours targets multimodality, theirs targets tail behavior).

## 3. Overview of Results

### 3.1. Results About Affine Coupling Architectures

We begin by proving several results for a particularly common normalizing flow architectures: those based on affine coupling layers (Dinh et al., 2014; 2016; Kingma & Dhariwal, 2018). The appeal of these architecture comes from training efficiency. Although layerwise invertible neural networks (i.e. networks for which each layer consists of an invertible matrix and invertible pointwise nonlinearity) seem like a natural choice, in practice these models have several disadvantages: for example, computing the determinant of the Jacobian is expensive unless the weight matrices are restricted.

Consequently, it's typical for the transformations in a flow network to be constrained in a manner that allows for efficient computation of the Jacobian determinant. The most common building block is an *affine coupling* block, originally proposed by (Dinh et al., 2014; 2016). A coupling block partitions the coordinates $[d]$ into two parts: $S$ and $[d] \setminus S$, for a subset $S$ with $|S|$ containing around half the coordinates of $d$. The transformation then has the form:

**Definition 1.** An *affine coupling block* is a map $f : \mathbb{R}^d \to \mathbb{R}^d$, s.t. $f(x_S, x_{[d] \setminus S}) = (x_S, x_{[d] \setminus S} \odot s(x_S) + t(x_S))$

Of course, the modeling power will be severely constrained if the coordinates in $S$ never change: so typically, flow models either change the set $S$ in a fixed or learned way (e.g. alternating between different partitions of the channel in (Dinh et al., 2016) or applying a learned permutation in (Kingma & Dhariwal, 2018)). As a permutation is a discrete object, it is difficult to learn in a differentiable manner – so (Kingma & Dhariwal, 2018) simply learns an invertible linear function (i.e. a 1x1 convolution) as a differentiation-friendly relaxation thereof.

#### 3.1.1. UNIVERSAL APPROXIMATION WITH ILL-CONDITIONED AFFINE COUPLING NETWORKS

First, we address universal approximation of normalizing flows and its close ties to conditioning. Namely, a recent work (Theorem 1 of (Huang et al., 2020)) showed that deep affine coupling networks are universal approximators if we allow the training data to be padded with sufficiently many zeros. While zero padding is convenient for their analysis (in fact, similar proofs have appeared for other invertible architectures like Augmented Neural ODEs (Zhang et al.)), in practice models trained on zero-padded data often perform poorly. Another work (Teshima et al., 2020) proves universal approximation with the optional permutations and $|S| = d - 1$ needed for the nonconstructive proof. We remove that requirement in two ways, first by giving a construction that gives universal approximation without permutations in 3 composed couplings and second by showing that the permutations can be simulated by a constant number of alternating but fixed coupling layers.

First we show that neither padding nor permutations nor depth is necessary representationally: shallow models without zero padding are already universal approximators in Wasserstein.

**Theorem 1** (Universal approximation without padding). *Suppose that $P$ is the standard Gaussian measure in $\mathbb{R}^n$ with $n$ even and $Q$ is a distribution on $\mathbb{R}^n$ with bounded support and absolutely continuous with respect to the Lebesgue measure. Then for any $\epsilon > 0$, there exists a depth-3 affine coupling network $g$, with maps $s, t$ represented by feedforward ReLU networks such that $W_2(g_\# P, Q) \leq \epsilon$.*

**Remark 1.** A shared caveat of the universality construction in Theorem 1 with the construction in (Huang et al., 2020) is that the resulting network is poorly conditioned. In the case

of the construction in (Huang et al., 2020), this is obvious because they pad the $d$-dimensional training data with $d$ additional zeros, and a network that takes as input a Gaussian distribution in $\mathbb{R}^{2d}$ (i.e. has full support) and outputs data on $d$-dimensional manifold (the space of zero padded data) must have a singular Jacobian almost everywhere.[1] In the case of Theorem 1, the condition number of the network blows up at least as quickly as $1/\epsilon$ as we take the approximation error $\epsilon \to 0$, so this network is also ill-conditioned if we are aiming for a very accurate approximation.

**Remark 2.** Based on Theorem 3, the condition number blowup of either the Jacobian or the Hessian is necessary for a shallow model to be universal, even when approximating well-conditioned linear maps. The network constructed in Theorem 1 is also consistent with the lower bound from Theorem 4, because the network we construct in Theorem 1 is highly non-Lipschitz and uses many parameters per layer.

3.1.2. THE EFFECT OF CHOICE OF PARTITION ON DEPTH

Next, we ask how much of a saving in terms of the depth of the network can one hope to gain from using learned partitions (ala GLOW) as compared to a fixed partition. More precisely:

**Question 1:** Can models like Glow (Kingma & Dhariwal, 2018) be simulated by a sequence of affine blocks with a fixed partition without increasing the depth by much?

We answer this question in the affirmative at least for equally sized partitions (which is what is typically used in practice). We show the following surprising fact: consider an arbitrary partition $(S, [2d] \setminus S)$ of $[2d]$, such that $S$ satisfies $|S| = d$, for $d \in \mathbb{N}$. Then for any invertible matrix $T \in \mathbb{R}^{2d \times 2d}$, the linear map $T : \mathbb{R}^{2d} \to \mathbb{R}^{2d}$ can be exactly represented by a composition of $O(1)$ affine coupling layers that are *linear*, namely have the form $L_i(x_S, x_{[2d] \setminus S}) = (x_S, B_i x_{[2d] \setminus S} + A_i x_S)$ or $L_i(x_S, x_{[2d] \setminus S}) = (C_i x_S + D_i x_{[2d] \setminus S}, x_{[2d] \setminus S})$ for matrices $A_i, B_i, C_i, D_i \in \mathbb{R}^{d \times d}$, s.t. each $B_i, C_i$ is diagonal. For convenience of notation, without loss of generality let $S = [d]$. Then, each of the layers $L_i$ is a matrix of the form $\begin{bmatrix} I & 0 \\ A_i & B_i \end{bmatrix}$ or $\begin{bmatrix} C_i & D_i \\ 0 & I \end{bmatrix}$, where the rows and columns are partitioned into blocks of size $d$.

With this notation in place, we show the following theorem:

**Theorem 2.** *For all $d \geq 4$, there exists a $k \leq 24$ such that for any invertible $T \in \mathbb{R}^{2d \times 2d}$ with $det(T) > 0$, there exist matrices $A_i, D_i \in \mathbb{R}^{d \times d}$ and diagonal matrices $B_i, C_i \in$*

---

[1]Alternatively, we could feed a degenerate Gaussian supported on a $d$-dimensional subspace into the network as input, but there is no way to train such a model using maximum-likelihood training, since the prior is degenerate.

$\mathbb{R}^{d \times d}_{\geq 0}$ *for all $i \in [k]$ such that*

$$T = \prod_{i=1}^{k} \begin{bmatrix} I & 0 \\ A_i & B_i \end{bmatrix} \begin{bmatrix} C_i & D_i \\ 0 & I \end{bmatrix}$$

Note that the condition $\det(T) > 0$ is required, since affine coupling networks are always orientation-preserving. Adding one diagonal layer with negative signs suffices to model general matrices. In particular, since permutation matrices are invertible, this means that any applications of permutations to achieve a different partition of the inputs (e.g. like in Glow (Kingma & Dhariwal, 2018)) can in principle be represented as a composition of not-too-many affine coupling layers, indicating that the flexibility in the choice of partition is not the representational bottleneck.

It's a reasonable to ask how optimal the $k \leq 24$ bound is – we supplement our upper bound with a lower bound, namely that $k \geq 3$. This is surprising, as naive parameter counting would suggest $k = 2$ might work. Namely, we show:

**Theorem 3.** *For all $d \geq 4$ and $k \leq 2$, there exists an invertible $T \in \mathbb{R}^{2d \times 2d}$ with $det(T) > 0$, s.t. for all $A_i, D_i \in \mathbb{R}^{d \times d}$ and for all diagonal matrices $B_i, C_i \in \mathbb{R}^{d \times d}_{\geq 0}, i \in [k]$ it holds that*

$$T \neq \prod_{i=1}^{k} \begin{bmatrix} I & 0 \\ A_i & B_i \end{bmatrix} \begin{bmatrix} C_i & D_i \\ 0 & I \end{bmatrix}$$

Beyond the relevance of this result in the context of how important the choice of partitions is, it also shows a lower bound on the depth for an equal number of *nonlinear* affine coupling layers (even with quite complex functions $s$ and $t$ in each layer) – since a nonlinear network can always be linearized about a (smooth) point to give a linear network with the same number of layers. In other words, studying linear affine coupling networks lets us prove a *depth lower bound/depth separation* for nonlinear networks for free.

**Remark 3** (Significance of Theorem 2 for Approximation in Likelihood/KL)**.** All of the universality results in the literature for normalizing flows, including Theorem 1, prove universality in the Wasserstein distance or in the related sense of convergence of distributions. A stronger and probably much more difficult problem is to prove universality under the KL divergence instead: i.e. to show for a well-behaved distribution $P$, there exists a sequence $Q_n$ of distributions generated by normalizing flow models such that

$$\mathrm{KL}(P, Q_n) \to 0. \tag{1}$$

This is important because Maximum-Likelihood training attempts to pick the model with the smallest KL, not the smallest Wasserstein distance, and the minimizers of these two objectives can be extremely different. For $P = N(0, \Sigma)$,

Theorem 2 certainly implies (1) for bounded depth linear affine couplings, and thus gives the first proof that global optimization of the max-likelihood objective of a normalizing flow model would successfully learn a Gaussian with arbitrary nondegenerate $\Sigma$.

### 3.2. Results about General Architectures

In order to guarantee that the network is invertible, normalizing flow models place significant restrictions on the architecture of the model. The most basic and general question we can ask is how this restriction affects the expressive power of the model — in particular, how much the depth must increase to compensate.

More precisely, we ask:

**Question 2:** is there a distribution over $\mathbb{R}^d$ which can be written as the pushforward of a Gaussian through a small, shallow generator, which cannot be approximated by the pushforward of a Gaussian through a small, shallow *layerwise invertible* neural network?

Given that there is great latitude in terms of the choice of layer architecture, while keeping the network invertible, the most general way to pose this question is to require each layer to be a function of $p$ parameters – i.e. $f = f_1 \circ f_2 \circ \cdots \circ f_\ell$ where $\circ$ denotes function composition and each $f_i : \mathbb{R}^d \to \mathbb{R}^d$ is an invertible function specified by a vector $\theta_i \in \mathbb{R}^p$ of parameters. This framing is extremely general: for instance it includes *layerwise invertible feedforward networks* in which $f_i(x) = \sigma^{\otimes d}(A_i x + b_i)$, $\sigma$ is invertible, $A_i \in \mathbb{R}^{d \times d}$ is invertible, $\theta_i = (A_i, b_i)$ and $p = d(d+1)$. It also includes popular architectures based on affine coupling blocks which we discussed in more detail in the previous subsection.

We answer this question in the affirmative: namely, we show for any $k$ that there is a distribution over $\mathbb{R}^d$ which can be expressed as the pushforward of a network with depth $O(1)$ and size $O(k)$ that cannot be (even very approximately) expressed as the pushforward of a Gaussian through a Lipschitz layerwise invertible network of depth smaller than $k/p$.

Towards formally stating the result, let $\theta = (\theta_1, \ldots, \theta_\ell) \in \Theta \subset \mathbb{R}^{d'}$ be the vector of all parameters (e.g. weights, biases) in the network, where $\theta_i \in \mathbb{R}^p$ are the parameters that correspond to layer $i$, and let $f_\theta : \mathbb{R}^d \to \mathbb{R}^d$ denote the resulting function. Define $R$ so that $\Theta$ is contained in the Euclidean ball of radius $R$.

We say the family $f_\theta$ is *L-Lipschitz with respect to its parameters and inputs*, if

$$\forall \theta, \theta' \in \Theta : \mathrm{E}_{x \sim \mathcal{N}(0, I_{d \times d})} \| f_\theta(x) - f_{\theta'}(x) \| \le L \| \theta - \theta' \|$$

and $\forall x, y \in \mathbb{R}^d, \| f_\theta(x) - f_\theta(y) \| \le L \| x - y \|$. [2] We will discuss the reasonable range for $L$ in terms of the weights after the Theorem statement. We show[3]:

**Theorem 4.** *For any* $k = \exp(o(d)), L = \exp(o(d)), R = \exp(o(d))$, *we have that for* $d$ *sufficiently large and any* $\gamma > 0$ *there exists a neural network* $g : \mathbb{R}^{d+1} \to \mathbb{R}^d$ *with* $O(k)$ *parameters and depth* $O(1)$, *s.t. for any family* $\{f_\theta, \theta \in \Theta\}$ *of layerwise invertible networks that are* $L$-*Lipschitz with respect to its parameters and inputs, have* $p$ *parameters per layer and depth at most* $k/p$ *we have*

$$\forall \theta \in \Theta, W_1((f_\theta)_{\#\mathcal{N}}, g_{\#\mathcal{N}}) \ge 10\gamma^2 d$$

*Furthermore, for all* $\theta \in \Theta$, $KL((f_\theta)_{\#\mathcal{N}}, g_{\#\mathcal{N}}) \ge 1/10$ *and* $KL(g_{\#\mathcal{N}}, (f_\theta)_{\#\mathcal{N}}) \ge \frac{10\gamma^2 d}{L^2}$.

**Remark 4.** First, note that while the number of parameters in both networks is comparable (i.e. it's $O(k)$), the invertible network is deeper, which usually is accompanied with algorithmic difficulties for training, due to vanishing and exploding gradients. For layerwise invertible generators, if we assume that the nonlinearity $\sigma$ is 1-Lipschitz and each matrix in the network has operator norm at most $M$, then a depth $\ell$ network will have $L = O(M^\ell)$[4] and $p = O(d^2)$. For an affine coupling network with $g, h$ parameterized by $H$-layer networks with $p/2$ parameters each, 1-Lipschitz activations and weights bounded by $M$ as above, we would similarly have $L = O(M^{\ell H})$.

**Remark 5.** We make a couple of comments on the "hard" distribution $g$ we construct, as well as the meaning of the parameter $\gamma$ and how to interpret the various lower bounds in the different metrics. The distribution $g$ for a given $\gamma$ will in fact be close to a mixture of $k$ Gaussians, each with mean on the sphere of radius $10\gamma^2 d$ and covariance matrix $\gamma^2 I_d$. Thus this distribution has most of it's mass in a sphere of radius $O(\gamma^2 d)$ — so the Wasserstein guarantee gives close to a trivial approximation for $g$. The KL divergence bounds are derived by so-called transport inequalities between KL and Wasserstein for subgaussian distributions (Bobkov & Götze, 1999). The discrepancy between the two KL divergences comes from the fact that the functions $g, f_\theta$ may have different Lipschitz constants, hence the tails of $g_{\#\mathcal{N}}$ and $f_{\#\mathcal{N}}$ behave differently. In fact, if the function $f_\theta$ had the same Lipschitz constant as $g$, both KL lower bounds would be on the order of a constant.

---

[2] Note for architectures having trainable biases in the input layer, these two notions of Lipschitzness should be expected to behave similarly.

[3] In this Theorem and throughout, we use the standard asymptotic notation $f(d) = o(g(d))$ to indicate that $\limsup_{d \to \infty} \frac{f(d)}{g(d)} = 0$. For example, the assumption $k, L, R = \exp(o(d))$ means that for any sequence $(k_d, L_d, R_d)_{d=1}^\infty$ such that $\limsup_{d \to \infty} \frac{\max(\log k_d, \log L_d, \log R_d)}{d} = 0$ the result holds true.

[4] Note, our theorem applies to exponentially large Lipschitz constants.

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
