# OpenReview forum: "Representational aspects of depth and conditioning in normalizing flows"
_ICML.cc/2021/Workshop/INNF — INNF+ 2021 spotlighttalk_

### Official Review · Reviewer_sxDg · 2021-06-04

**Rating:** Accept
**Confidence:** 3

**Summary:**

The paper investigates several important problems in theoretically understanding normalizing flow models. In particular, it claims to prove that shallow affine coupling flows without zero padding are enough for approximating well-behaved distribution in Wasserstein distance, and learned partition used in Glow can be approximated composing a small number of fixed partitions with affine coupling.

**Justification For Rating:**

The problems being addressed in this work are important to the normalizing flow community. Compared to previous papers on the expressivity of NFs, the theorems in this paper make fewer assumptions and the settings are closer to what is used in practice (zero padding, depth etc.). Of course, it is difficult to fit a theoretical work into 4 pages, so it is important to point to the full version of the paper.

---

### Official Review · Reviewer_wHph · 2021-06-12

**Rating:** Accept
**Confidence:** 3

**Summary:**

The work studies the representation power of normalizing flows on two important aspects, depth and conditioning, in a theoretical approach. The major results from the theoretical study are
1. For normalizing flow built with affine coupling layers, shallow models without padding is a universal approximator in Wasserstein sense in spite of being ill-conditioned.
2. Invertible linear transformations can be exactly replaced with composition of linear affine coupling layers.
3. There exists distributions that can be represented as a pushforward by a non-invertible shallow network and can not be represented as a pushforward by a shallow normalizing flow.

**Justification For Rating:**

The work provides important clarification and insights into fundamental questions about the representation power of normalizing flows and its relation with model depth and conditioning number. The theoretical results from the paper would be more solid if proof of the theorems can be presented in the appendix.

However, empirical study accompanying theoretical results is critical to understand the challenges of applying the theoretical results mentioned in the remarks. For example, the challenge of ill-conditioning due to accurate approximation with shallow network in Remark 1 might be better illustrated with some toy examples; fitting the “hard” distribution g in Remark 5 can also be studied and demonstrated with experiments on synthetic data. Empirical study results might also help bridging the gap between approximation in Wasserstein distance sense and approximation in KL sense as maximum likelihood training is usually used to optimize flow models.

The reviewer is looking forward to seeing further expansion of interesting workshop paper into a full-length paper. The contributions of the work itself can justify its acceptance.

---

### Decision · Program_Chairs · 2021-06-14

Accept (spotlight talk)